# *TERC* Variants Associated with Short Leukocyte Telomeres: Implication of Higher Early Life Leukocyte Telomere Attrition as Assessed by the Blood-and-Muscle Model

**DOI:** 10.3390/cells9061360

**Published:** 2020-05-31

**Authors:** Simon Toupance, Maria G. Stathopoulou, Alexandros M. Petrelis, Vesna Gorenjak, Carlos Labat, Tsung-Po Lai, Sophie Visvikis-Siest, Athanase Benetos

**Affiliations:** 1Université de Lorraine, Inserm, DCAC, F-54000 Nancy, France; s.toupance@chru-nancy.fr (S.T.); carlos.labat@inserm.fr (C.L.); 2Université de Lorraine, CHRU-Nancy, Pôle “Maladies du Vieillissement, Gérontologie et Soins Palliatifs”, F-54000 Nancy, France; 3Université de Lorraine, IGE-PCV, F-54000 Nancy, France; maria.stathopoulou@inserm.fr (M.G.S.); apetrelis@live.com (A.M.P.); gorenjak.vesna@gmail.com (V.G.); sophie.visvikis-siest@inserm.fr (S.V.-S.); 4Center of Human Development and Aging, Rutgers, New Jersey Medical School, The State University of New Jersey, Newark, NJ 07103, USA; tl599@njms.rutgers.edu

**Keywords:** telomere length maintenance, SNPs, TERC, atherosclerosis, cohort studies

## Abstract

Short leukocyte telomere length (LTL) is associated with atherosclerotic cardiovascular disease (ASCVD). Mendelian randomisation studies, using single nucleotide polymorphisms (SNPs) associated with short LTL, infer a causal role of LTL in ASCVD. Recent results, using the blood-and-muscle model, indicate that higher early life LTL attrition, as estimated by the ratio between LTL and skeletal muscle telomere length (MTL), rather than short LTL at conception, as estimated by MTL, should be responsible of the ASCVD-LTL connection. We combined LTL and MTL measurements and SNPs profiling in 402 individuals to determine if 15 SNPs classically described as associated with short LTL at adult age were rather responsible for higher LTL attrition during early life than for shorter LTL at birth. Two of these SNPs (rs12696304 and rs10936599) were associated with LTL in our cohort (*p* = 0.027 and *p* = 0.025, respectively). These SNPs, both located on the *TERC* gene, were associated with the LTL/MTL ratio (*p* = 0.007 and *p* = 0.037, respectively), but not with MTL (*p* = 0.78 and *p* = 0.32 respectively). These results suggest that SNPs located on genes coding for telomere maintenance proteins may contribute to a higher LTL attrition during the highly replicative first years of life and have an impact later on the development of ASCVD.

## 1. Introduction

Short leukocyte telomere length (LTL) is associated with atherosclerotic cardiovascular disease (ASCVD) and its complications [1,2]. Genome-wide association studies (GWAS) have found several single nucleotide polymorphisms (SNPs) associated with short LTL over the years [3,4] and recently Mendelian randomisation studies showed that these genetic variants are associated with ASCVD [5,6,7] inferring a causal role of short LTL in atherosclerosis. The hypothesis is that inherited short LTL is associated with ASCVD. However, we recently showed, using the blood and muscle model, that the association between short LTL and ASCVD was not related to shorter telomere length (TL) at the beginning of life—as estimated by minimally proliferative skeletal muscle telomere length (MTL)—but to higher LTL attrition, especially during the first years of life, as estimated by the LTL/MTL ratio [8]. This lead us to the hypothesis that genetic variants associated with short LTL measured at adult age identified by GWAS could be related to higher LTL attrition during early life rather than to shorter LTL at conception. In this genetic study, we tested this hypothesis by combining SNPs profiling of 15 variants associated with short LTL and telomere length measurements in leukocytes and skeletal muscles. We found that SNPs associated with short LTL in our cohort, *TERC* variants rs12696304 and rs10936599, are linked with higher estimated early life LTL attrition rather than with shorter estimated LTL at birth.

## 2. Materials and Methods

### 2.1. The Cohort

The aim of the TELARTA (Telomere in Arterial Aging) study is to examine the implication of telomere dynamics in arterial aging and the development of atherosclerosis using the blood and muscle model. The study and its goals have been described previously [8]. Briefly, 259 men and women (older than 20 years), who were admitted for various surgical procedures, were enrolled in university hospitals in Nancy (*n* = 215) and Marseille (*n* = 44), France. A replication cohort of 143 individuals was enrolled in 3 sites: 91 individuals in original sites (Nancy and Marseille) and 52 individuals in 3 Athens hospitals (Onassis Cardiac Surgery Center, Surgeon KP; Iaso General Hospital Surgeon MVG; Hippokration Hospital, Surgeon EM). All 402 subjects gave their informed consent for inclusion before they participated in the study. The study was conducted in accordance with the Declaration of Helsinki, and the protocol was approved by Ethics Committee (Comité de Protection des Personnes) of Nancy, France, Ethics Committee of the University of Athens and Ethics Committee of each one of the 3 participating hospitals. This study is registered on http://www.clinicaltrials.gov under unique identifier: NCT02176941.

### 2.2. Telomere Length Measurements

Leukocyte TL (LTL) and Muscle TL (MTL) were measured in DNA extracted by the phenol/chloroform/isoamyl alcohol method from peripheral blood leukocytes and skeletal muscle biopsies (~ 100 mg in the surgical field), respectively. DNA samples passed an integrity testing using a 1% (wt/vol) agarose gel before TL measurements performed by Southern blotting of the terminal restriction fragments (TRFs), as described previously [9]. Briefly, DNA samples were digested (37 °C) overnight with the restriction enzymes Hinf I and Rsa I (Roche Diagnostics GmbH, Mannheim, Germany). Digested DNA samples and DNA ladders were resolved on 0.5% (wt/vol) agarose gels during 23 h. DNA was depurinated, denatured, neutralised and transferred onto a positively charged nylon membrane (Roche Diagnostics GmbH) using a vacuum blotter (Biorad, Hercules, CA, USA). Membranes were hybridised at 65 °C with a DIG-labelled telomeric probe after which the probe was detected by the DIG luminescent detection procedure (Roche Diagnostics GmbH) and exposed on CCD camera (Las 4000, Fujifilm Life Sciences, Cambridge, MA, USA). Optical density values (OD) versus DNA migration distances (in pixels) were obtained from pictures with image processing software (Multigauge, Fujifilm Life Sciences). Distances were converted to molecular weight (MW) in kilobases (kb) using a power function transformation owing to DNA ladders. Mean TRF length was calculated in the 3–20 kb range using the formula mean TRF (kb) = Σ (ODi)/Σ (ODi/MWi), where ODi is the OD at position i and MWi is the MW (i.e., TRF length) at that position. Leukocyte and muscle DNA from one patient were always measured simultaneously on the same membrane, as shown in the representative blot (Figure 1), and measurements were performed in duplicate on separate membranes. The measurement repeatability, as determined by the intraclass correlation coefficient, was 0.99 (95% confidence interval, 0.817–1.0) and 0.98 (95% confidence interval, 0.81–1.0) for LTL and MTL, respectively. The repeatability of the means of 2 duplicates (used in the analysis), known as the extrapolated repeatability, was 0.995 and 0.991 for LTL and MTL, respectively. The LTL/MTL ratio was calculated.

### 2.3. Single Nucleotide Polymorphism Genotyping

Fifteen SNPs (rs11125529, rs6772228, rs12696304, rs10936599, rs7675998, rs7726159, rs2736100, rs9419958, rs9420907, rs4387287, rs3027234, rs8105767, rs412658, rs6028466 and rs755017), observed previously to be associated with LTL in several GWAS [5], were genotyped in DNA samples using a PCR-based KASP assay [10]. Minor allele frequencies (MAF) were calculated and the Hardy-Weinberg equilibrium was tested using the χ^2^ method. SNPs which did not follow Hardy-Weinberg equilibrium were excluded in the subsequent analyses. 

### 2.4. Statistical Analyses

Continuous variables are presented as means ± SD or mean ± SE and discrete variables as frequencies or percentages. Pairwise comparisons were performed using the Mann–Whitney and χ^2^ tests, as appropriate. Telomere length values are presented and compared with or without adjustment to age and sex. Bivariate relations between continuous variables were determined using Pearson correlation coefficients. Statistical analyses were performed using the Number Crunching Statistical System (NCSS) 9 statistical software package (NCSS, Kaysville, UT, USA).

For SNP analyses, LTL was transformed into logLTL to normalize its distribution. Statistical analyses were performed using linear regression models adjusted for age and sex for LTL and MTL, and age only for the LTL/MTL ratio (MTL and LTL but not LTL/MTL are sex-dependant). Direct SNP associations were examined with PLINK software in an additive model (using minor alleles as reference alleles) and epistatic interactions with PLINK and R softwares and the CAPE R package. Taking into account the limited sample size, and in order to limit multiple testing, the LTL phenotype was initially tested and only SNPs associated with LTL in our cohort were tested against MTL and LTL/MTL phenotypes. 

## 3. Results

### 3.1. Population Characteristics

Participants were of European ancestry, aged 60 ± 15 years (with an age range of 20–94 years) and 32% were females. Among them, 48% displayed clinical manifestations of ASCVD in either coronary, carotid, cerebral, iliac, femoral or popliteal arteries.

### 3.2. Telomere Length Dynamics

LTL (mean ± SD; 6.71 ± 0.84 kb) was shorter than MTL (8.57 ± 0.72 kb; *p* < 0.0001) in all individuals. Mean LTL/MTL ratio was 0.78 ± 0.07. Both LTL and MTL shortened with age with a slope of the effect of age steeper for LTL than MTL (−31 bp/year and −15 bp/year, respectively). The LTL/MTL ratio decreased with age (−0.0021 ± 0.0002/year). Age-adjusted LTL was longer in women than men (6.86 ± 0.06 kb and 6.64 ± 0.04 kb, respectively; *p* < 0.005). Similarly, age-adjusted MTL was longer in women than men (8.74 ± 0.06 kb and 8.49 ± 0.04 kb, respectively; *p* < 0.001), whereas age-adjusted LTL/MTL was not influenced by sex (0.78 ± 0.005 in women vs. 0.78 ± 0.004 in men; *p* = 0.69). 

### 3.3. Single Nucleotide Polymorphisms Association with Telomere Length

The 13 SNPs in accordance with Hardy–Weinberg equilibrium (rs11125529, rs6772228, rs12696304, rs10936599, rs7675998, rs2736100, rs9419958, rs9420907, rs4387287, rs3027234, rs8105767, rs412658 and rs6028466) were included in the analyses. Two of these SNPs, rs12696304 and rs10936599, were significantly associated with LTL (*p* = 0.027 and *p* = 0.025, respectively, Table 1). These SNPs map to a locus harbouring *TERC*, the gene encoding the RNA subunit of telomerase. The minor alleles, allelic frequencies, and directions of the associations were similar to previous findings [5]. These two SNPs showed no association with MTL (*p* = 0.78 and *p* = 0.32, respectively, Table 2) but were associated with the LTL/MTL ratio (*p* = 0.007 and *p* = 0.037, respectively), in a way that the allele associated with shorter LTL was associated with a lower LTL/MTL ratio, thus signalling higher LTL attrition (Figure 2). No significant epistatic interaction was identified.

## 4. Discussion

In our cohort of patients, SNPs that were found to be associated with short LTL are associated with a lower LTL/MTL ratio but are not associated with MTL. The present results indicate that rs12696304 and rs10936599, two *TERC* SNPs previously described as associated with short LTL, do not influence TL at conception as reflected by MTL, but rather higher LTL attrition from conception onwards as expressed in the LTL/MTL ratio. We have reported previously that the gap between LTL and MTL is essentially constituted during the first decade of life [11]. SNPs located on genes coding for telomere maintenance proteins may thus contribute to higher TL attrition during this critical highly replicative period and have an impact later on the development of ASCVD [12,13,14].

The observed associations of these SNPs with LTL but not MTL may also suggest that person-to-person variation in TL dynamics specifically in the hematopoietic system but not in skeletal muscle, and presumably not in most other tissues, might contribute to the pathogenesis of atherosclerosis. This contribution may possibly be mediated through inflammation regulation [15]. Indeed, short telomeres have been shown to upregulate the NLRP3 (NOD (nucleotide oligomerization domain)-, LRR (leucine-rich repeat)-, and PYD (pyrin domain)-containing protein 3) inflammasome of macrophages in a murine model [16]. NLRP3, an innate immune signalling complex, is known to activate the proinflammatory IL-1 pathway, which contribute to atherogenesis [17]. Moreover, clonal haematopoiesis of indeterminate potential (CHIP), a phenomenon which is associated with a doubling of the risk of ASCVD [18,19], has been recently linked to short LTL [20,21] and the upregulation of the macrophage NLRP3 inflammasome in human [22,23]. Another possible explanation of the link observed between short LTL and ASCVD is through the hemothelium paradigm. As the hematopoietic system and vascular endothelium share a common precursor, the hemogenic endothelium or hemothelium [24], short LTL might reflect short TL in progenitor cells involved in endothelial repair. Compromised proliferation capacity of endothelial progenitor cells with short TL may influence endothelium repair capacity and impact ASCVD progression [25].

This study is limited by the small sample size in regard to SNPs analyses, but it is difficult to obtain very large number of participants due to the heavy procedure needed to obtain results for MTL. However, this drawback was offset at least for TL measurements by the use of high reproducibility TRF method [26,27].

In conclusion, taking advantage from the data obtained by the blood and muscle telomere model and those obtained from Mendelian randomisation studies, the present study proposes a new mechanical insight in the association between short LTL and ASCVD. Our findings suggest that *TERC* variants might impact early life TL attrition in hematopoietic cells and contribute later to the pathogenesis of atherosclerosis.

## Figures and Tables

**Figure 1 cells-09-01360-f001:**
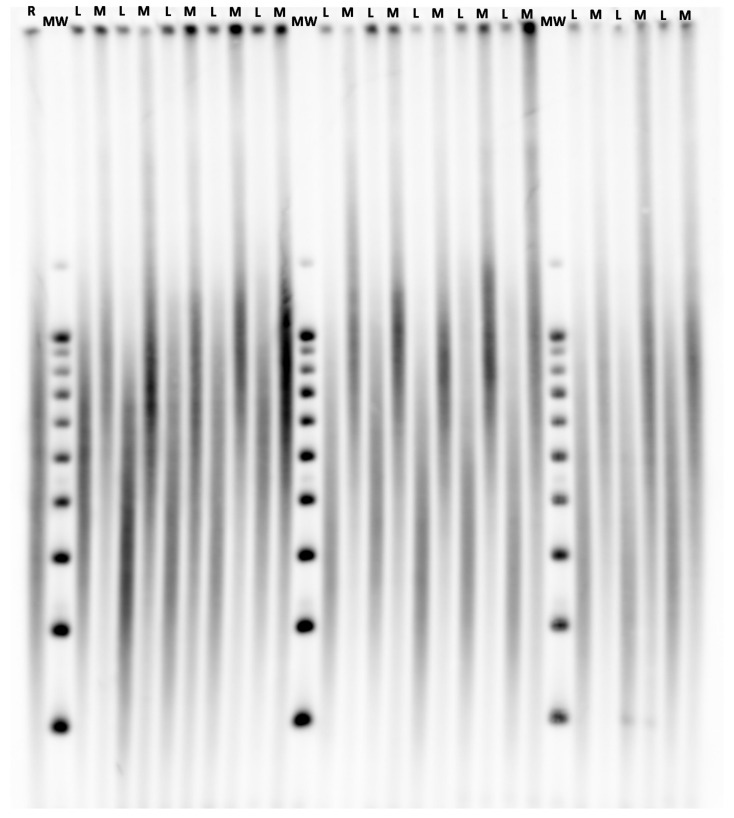
Illustrative membrane of telomere length measurements performed by Southern blotting of the terminal restriction fragments (TRFs). A sample of known telomere length serves as an internal reference (**R**). Three molecular weight ladders (**MW**) are resolved across the gel, and the one that is closest to a given sample is used for the computation of its mean TRF. Leukocyte (**L**) and Muscle (**M**) DNA samples of one participant are resolved on adjacent lanes.

**Figure 2 cells-09-01360-f002:**
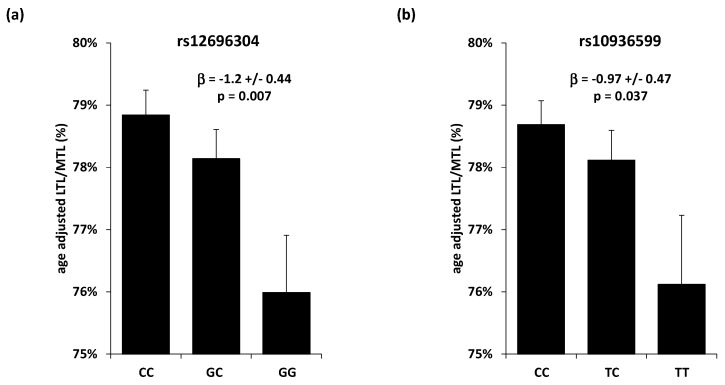
*TERC* SNPs association with LTL/MTL (**a**) LTL/MTL ratio (expressed in %) according to Table 12696304. (*TERC*) genotype; (**b**) LTL/MTL ratio (expressed in %) according to the rs10936599 (*TERC*) genotype. LTL = Leukocyte telomere length, MTL = Muscle telomere length, β = change in ratio per copy of the effect allele Values are presented as mean +/− SEM. Analyses are age-adjusted.

**Table 1 cells-09-01360-t001:** Single nucleotide polymorphisms association with leukocyte telomere length (LTL).

							Log LTL		
SNP	CHR	Position	Gene	Effect Allele	Other Allele	MAF	BETA	SE	P
rs11125529	2	54248729	*ACYP2*	A	C	0.12	0.000024	0.0050	0.99
rs6772228	3	58390292	*PXK*	A	T	0.045	−0.0094	0.0073	0.20
rs12696304	3	169763483	*TERC*	G	C	0.29	−0.0074	0.0033	0.027
rs10936599	3	169774313	*TERC*	T	C	0.25	−0.0080	0.0036	0.025
rs7675998	4	163086668	*NAF1*	A	G	0.24	−0.0012	0.0037	0.74
rs2736100	5	1286401	*TERT*	A	C	0.47	−0.0038	0.0032	0.24
rs9419958	10	103916188	*OBFC1*	T	C	0.16	−0.00093	0.0044	0.84
rs9420907	10	103916707	*OBFC1*	C	A	0.17	0.00032	0.0042	0.94
rs4387287	10	103918139	*OBFC1*	A	C	0.21	0.0038	0.0039	0.32
rs3027234	17	8232774	*CTC1*	T	C	0.21	0.0017	0.0038	0.66
rs8105767	19	22032639	*ZNF208*	G	A	0.30	−0.0027	0.0034	0.42
rs412658	19	22176638	*ZNF676*	T	C	0.39	−0.0016	0.0032	0.63
rs6028466	20	39500359	*DHX35*	A	G	0.070	−0.0022	0.0062	0.72

LTL = Leukocyte telomere length, SNP = Single nucleotide polymorphism, CHR = Chromosome, Position = base-pair position (GRCh38.p3), MAF = minor allelic frequency, BETA = regression coefficient (change in telomere length per copy of the effect allele), SE = standard error. LTL values are transformed in Log LTL to normalize their distribution. Analyses are age- and sex-adjusted.

**Table 2 cells-09-01360-t002:** TERC single nucleotide polymorphisms (SNPs) association with muscle telomere length (MTL).

						MTL		
SNP	CHR	Position	Gene	Effect Allele	Other Allele	BETA	SE	P
rs12696304	3	169763483	*TERC*	G	C	−0.014	0.051	0.78
rs10936599	3	169774313	*TERC*	T	C	−0.054	0.054	0.32

MTL = Muscle telomere length, SNP = Single nucleotide polymorphism, CHR = Chromosome, Position = base-pair position (GRCh38.p3), BETA = regression coefficient (change in telomere length per copy of the effect allele), SE = standard error. Analyses are age- and sex-adjusted.

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
