# Peer review of "TERC Variants Associated with Short Leukocyte Telomeres: Implication of Higher Early Life Leukocyte Telomere Attrition as Assessed by the Blood-and-Muscle Model"

_cells, 2020, doi:10.3390/cells9061360_

Round 1
Reviewer 1 Report
The manuscript by Toupance et al., entitled “TERC Variants Associated with Short Leukocyte Telomeres: Implication of Higher Early Life Leukocyte Telomere Attrition as Assessed by the Blood-and-Muscle Model”
points out that specific SNPs located on TERC gene, coding for telomere proteins, may be involved in LTL attrition early in life and later on in ASCVD development. The study further explores the effectiveness of Blood-and-Muscle Model, evaluating association to LTL of several single nucleotide polymorphism. In my opinion, the topic is interesting, the manuscript is well written and conclusions are coherent with the initial hypothesis. However, there are some minor points that need to be clarified:
- What induced the authors to use the TRF technique, quite labor intensive, time consuming and less accurate of new real time approaches? It would be appreciable to refer to modern technique, at least in the perspective of future studies.
- It is not clear how many patients of the described population for LTL-SNPs analysis were involved also in MTL analysis, all of them or a subpopulation? Is “small sample size” (line 167) referred to MTL analysis only? Please specify more in detail.
- Line 85: please correct to “nucleotide”
Author Response
- What induced the authors to use the TRF technique, quite labor intensive, time consuming and less accurate of new real time approaches? It would be appreciable to refer to modern technique, at least in the perspective of future studies.
As we stated in the text, it is quite difficult to obtain a large number of subjects for this type of analyses since it requires skeletal muscle tissue which is more difficult to obtain than blood samples. Therefore, we choose to use the most accurate method to measure TL. TRF is labor intensive and time consuming but it is the most reproducible method and the best fit for small population analyses (Aviv et al, Nucleic Acids Res. 2011, Nettle et al, PLoS One 2019). We added a sentence in the Discussion with those references (lines 188-189).
- It is not clear how many patients of the described population for LTL-SNPs analysis were involved also in MTL analysis, all of them or a subpopulation? Is “small sample size” (line 167) referred to MTL analysis only? Please specify more in detail.
All the 402 subjects included in the analyses had LTL, MTL and SNPs measurements. The “small sample size” is in regard of SNP profiling. While 402 subjects can be considered as a sufficient number for TL dynamics analyses using the TRF method, usually SNPs association studies uses samples size of thousands more than hundreds. We clarified in the Discussion (lines 186-189).
- Line 85: please correct to “nucleotide”
It has been corrected.
Reviewer 2 Report
In this work , the authors analyzed several variants associated with short LTL and telomere length measurements in leukocytes and skeletal muscles. Interestingly, they found that SNPs associated with short LTL, TERC variants rs12696304 and rs10936599, were linked with higher estimated early life LTL attrition rather than with shorter estimated LTL at birth.
This paper is well-well written and structured. I have some questions:
-Are the enrolled subjects 402 (as reported in the Study Cohort) or 403 (as reported in the Abstract)?
-What was the quality of DNA for TL analysis? What was the inter-assay coefficient?
- It could be interesting to analyze the combined effects of the risk alleles of two SNPs in TERC gene.
Author Response
-Are the enrolled subjects 402 (as reported in the Study Cohort) or 403 (as reported in the Abstract)?
The enrolled subjects are 402 as reported in the Methods section. It has been corrected in the Abstract.
-What was the quality of DNA for TL analysis? What was the inter-assay coefficient?
As stated in the methods, DNA integrity was controlled in 1% agarose gel. DNA with signs of degradation (shift downward of the DNA “crown” or smear under the crown) were considered unfit for the analyses. The inter-assay coefficient of variation for duplicate measurements on different membranes was 1.2%. In the manuscript, the reproducibility of TL measurements was assessed by intraclass correlation coefficient rather than with coefficient of variation.
- It could be interesting to analyze the combined effects of the risk alleles of two SNPs in TERC gene.
Epistatic interactions have been tested and we did not identify any significant interaction between these two SNPs.
Reviewer 3 Report
In this manuscript entitled "TERC Variants Associated with Short Leukocyte Telomeres: Implication of Higher Early Life Leukocyte Telomere Attrition as Assessed by the Blood-and-Muscle Model” by Toupance et. al. examined telomere lengths in blood and muscle, and the association with atherosclerotic cardiovascular disease (ASCVD). Upon careful examination, I suggest the following revisions:
1) The RNA structural domains of TERC has been well documented. The authors should show the SNPs in the context of TERC domains and discuss how these SNPs might have cause the functional alteration of TERC.
2) The telomere lengths were measured by TRF. The authors should show representative southern blot images. Also, the authors should specify how exactly they derived the mean TRF lengths from the “smears” on TRF gels, assuming a single telomere length readout from each sample was used for the statistics.
3) Dot plots showing LTL/MTL and population characteristics would be helpful.
Author Response
1) The RNA structural domains of TERC has been well documented. The authors should show the SNPs in the context of TERC domains and discuss how these SNPs might have cause the functional alteration of TERC.
The two polymorphisms rs12696304 and rs10936599 are located near but not inside the genetic locus of TERC gene. Thus, they do not directly affect the RNA structure of TERC. They are located in regions of regulatory elements for gene expression but their biological mechanism on TERC functionality is not known.
2) The telomere lengths were measured by TRF. The authors should show representative southern blot images. Also, the authors should specify how exactly they derived the mean TRF lengths from the “smears” on TRF gels, assuming a single telomere length readout from each sample was used for the statistics.
We added a figure with a representative blot (Figure 1). In the Methods section, we completed the paragraph to explain how the mean TRF was calculated from raw data (lines 75-80).
3) Dot plots showing LTL/MTL and population characteristics would be helpful.
We added the figure showing LTL/MTL ratios according to the genotypes (Figure 2). Since the distribution of the values follows normal distribution and adjustment to age was applied, we presented these data as histograms rather than dot plots.